# Identification of Genomic Regions Controlling Chalkiness and Grain Characteristics in a Recombinant Inbred Line Rice Population Based on High-Throughput SNP Markers

**DOI:** 10.3390/genes12111690

**Published:** 2021-10-24

**Authors:** Yheni Dwiningsih, Anuj Kumar, Julie Thomas, Charles Ruiz, Jawaher Alkahtani, Abdulrahman Al-hashimi, Andy Pereira

**Affiliations:** 1Department of Crop, Soil and Environmental Sciences, University of Arkansas, Fayetteville, AR 72701, USA; ydwining@uark.edu (Y.D.); axk018@uark.edu (A.K.); jt008@uark.edu (J.T.); ceruiz@uark.edu (C.R.); 2Department of Botany and Microbiology, College of Science, King Saud University, Riyadh 11451, Saudi Arabia; Jsalqahtani@ksu.edu.sa (J.A.); Aalhashimi@ksu.edu.sa (A.A.-h.)

**Keywords:** rice, chalkiness, SNP, QTL, gene

## Abstract

Rice (*Oryza sativa* L.) is the primary food for half of the global population. Recently, there has been increasing concern in the rice industry regarding the eating and milling quality of rice. This study was conducted to identify genetic information for grain characteristics using a recombinant inbred line (RIL) population from a japonica/indica cross based on high-throughput SNP markers and to provide a strategy for improving rice quality. The RIL population used was derived from a cross of “Kaybonnet (KBNT *lpa*)” and “ZHE733” named the K/Z RIL population, consisting of 198 lines. A total of 4133 SNP markers were used to identify quantitative trait loci (QTLs) with higher resolution and to identify more accurate candidate genes. The characteristics measured included grain length (GL), grain width (GW), grain length to width ratio (RGLW), hundred grain weight (HGW), and percent chalkiness (PC). QTL analysis was performed using QTL IciMapping software. Continuous distributions and transgressive segregations of all the traits were observed, suggesting that the traits were quantitatively inherited. A total of twenty-eight QTLs and ninety-two candidate genes related to rice grain characteristics were identified. This genetic information is important to develop rice varieties of high quality.

## 1. Introduction

Rice (*Oryza sativa* L.) is the major dietary component for approximately more than half of the global population [1], and the demand for high-quality rice is gradually increasing due to the increasing standards of living [2]. The indicators of high-quality rice are based on physical characteristics and sensory properties such as milling performance, grain appearance, aroma, cooking, and nutrient composition [3]. These indicators are becoming important factors in rice marketing, which affects consumer acceptance of rice. Grain appearance quality is determined by grain length (GL), grain width (GW), grain length to width ratio (RGLW), hundred grain weight (HGW), and percent chalkiness (PC) or translucency [4]. Preference of rice quality is also influenced by cultures and habits [5]. Slender and long grains having an RGLW of three or above are generally preferred by the majority of rice consumers in most Asian countries, including India, China, and Thailand. Meanwhile, short and wide grains are preferred in Japan and Korea [6]. Thus, rice breeders have a challenging problem to improve rice grain quality and yield and need to consider the appropriate grain shape for market requirements.

Grain chalkiness has become an important problem in many global rice-producing areas since chalky grains negatively affect the cooking, eating, grain appearance, and milling of rice [4,7,8,9]. Chalkiness is an opaque area that occurs at the center, belly, or back of or throughout the grain and can occupy more than 50% of the total area [10,11,12,13]. Rice markets will not accept rice with more than 2% PC. PC is related to the content and structure of starch granules that contain amylose and amylopectin in the endosperm. Amylose biosynthesis is catalyzed by ADP-glucose pyrophosphorylase (AGPase) and granule-bound starch synthase (GBSS) in the endosperm, while amylopectin is synthesized by starch biosynthetic isozymes including AGPase, starch synthases (SSs), starch branching enzymes (BEs), and starch debranching enzymes (DBEs) [14]. Chalky grains occur when rice is grown at a high temperature, particularly under higher night temperature (HNT) conditions during grain filling stages [15,16,17,18]. The percentage of chalkiness under HNT conditions also depends on the rice variety [18]. According to Yamakawa et al. (2007) [16], HNT caused downregulation of amylose and amylopectin biosynthesis. According to Del Rosario et al. (1968) [19], chalky grains are prone to breakage during milling because of the lower density of starch granules compared to translucent grains. Broken rice grains sell at an almost 50% lower price compared to unbroken rice [20]. In addition, chalky grains also reduce the palatability of cooked rice due to the transverse and longitudinal cracks in the chalky area [21,22,23]. In the USA, as the third largest exporter of rice, long grain rice varieties are dominant, but they are susceptible to breakage due to chalkiness [20]; thus, it is crucial to enhance our understanding to improve the rice quality in competitive commercial markets. 

Scanning electron microscopy (SEM) analysis has shown different starch granule morphologies between chalky and translucent rice grains that significantly affected their textural, cooking, and physicochemical properties [8,9]. Chalky grains have starch granules that are loosely packed with greater air spaces between them, with a disordered cellular structure, rounded appearance, and large compounds, whereas translucent grains showed a tightly packed, polyhedral shape and small single starch granules [16,24,25,26,27]. Chalky grains displayed an opaque appearance and were prone to breaking during milling due to the large air spaces between the starch granules that hinder light transmission and create instability within the cellular structure [28].

Rice grain appearance characteristics, including GL, GW, RGLW, HGW, and PC, have been confirmed as quantitative traits, and numerous quantitative trait loci (QTLs) controlling grain appearance were reported [29,30,31,32,33,34,35,36,37,38,39,40,41,42,43,44,45,46]. Edwards et al. (2017) [33] suggested that a low phytic acid locus on chromosome 2 may be one cause of grain chalkiness. In 2019, Chen et al. [31] validated a QTL for grain width (*qGW2*) on chromosome 2 that might influence chalkiness by using a high-throughput SNP marker. *GW2* on chromosome 2, a QTL associated with grain width and weight, encodes a RING-type protein E3 ubiquitin ligase activity, and loss of *GW2* function results in increased grain width, weight, and yield [45]. *GS3* on chromosome 3 was identified as a major QTL for grain weight and length and a minor QTL for grain width and thickness that encodes a putative transmembrane protein, and also as a negative regulator for grain size [46]. Hu et al. (2018) [32] identified an important QTL, *qTGW3*, on chromosome 3 that negatively regulates grain length and weight, encoding GSK3/SHAGGY-like kinase OsGSK5/OsSK41, and interacts with OsARF4. A minor QTL associated with small grain size on chromosome 3 (*SG3*) encoding an R2R3-MYB protein that negatively regulates grain length was detected [30]. On chromosome 5, the gene *qSW5/GW5* related to grain shape was cloned by Shomura et al. (2008) [44] and Weng et al. (2008) [43], and a deletion in this gene was associated with significantly increased grain width. In the adjacent region of *qSW5/GW5*, *GS5* on chromosome 5 encodes a putative serine carboxypeptidase as a positive governor of grain size [41]. *Chalk5*, a major QTL for grain chalkiness that encodes a vacuolar H+-translocating pyrophosphatase, was identified by Li et al. (2014) [36]. *G46B* on chromosome 5, a major QTL associated with increasing grain width, grain thickness, 1000 grain weight, and chalkiness, was detected by using a recombinant inbred line population [34]. The gene *TGW6* regulating thousand grain weight, which encodes an indole-3-acetic acid (IAA)-glucose hydrolase activity, was cloned by Ishimaru et al. (2013) [37], and loss of function of this gene increases rice grain weight and yield. 

The grain chalkiness QTL *qPGWC-7* on chromosome 7, containing 13 candidate genes, was defined by Zhao et al. (2009) [42]. A fine mapped QTL associated with grain size was also identified on chromosome 7, *GS7/qSS7* [39,40]. *GW8* (OsSPL16) on chromosome 8 is a positive regulator of grain width and yield [38]. Another QTL for grain chalkiness on chromosome 8, *qPGC8-2*, was validated by Sun et al. (2015) [35]. A novel QTL for grain length and 1000 grain weight (*qGL11*) on chromosome 11 was validated by using chromosome segment substitution lines [29]. In 2019, Chen et al. [31] also validated a QTL for grain length (*qGL12*) on chromosome 12 by using a high-throughput SNP marker. Therefore, it is important to identify novel QTLs or candidate genes associated with the chalkiness and grain characteristics for the improvement of grain quality. The objective of this research was to identify the genetic information on rice grain characteristics using a recombinant inbred line population from a japonica/indica cross based on high-throughput SNP markers and to provide a strategy for improving rice quality. According to our previous study of chalkiness and grain characteristics using diverse rice genotypes, we found that the upland japonica variety Kaybonnet and the indica variety ZHE733 displayed a significant difference in percent chalkiness. 

## 2. Materials and Methods

### 2.1. Plant Material

A recombinant inbred line (RIL) population consisting of 198 lines, derived from the cross of the upland japonica variety Kaybonnet (KBNT *lpa*) and the indica variety ZHE733 (named K/Z RIL population), and the two parents were planted in a randomized complete block design with five replications for each parent and line in the growing season (May-November) in 2018 at the Agricultural Research Center, Fayetteville, AR, USA. A total of two parents and 198 lines with five replications for each parent and line were used for experiments and data analysis. Field management followed the normal agronomic procedures. The K/Z RIL population is a registered mapping population by Rutger and Tai (2005) [47], and it was generated from an F2 population by single seed descent (SSD) after selfing for F10 generations (USDA, Stuttgart, AR, USA). This population was chosen for the study because there is a significant difference between the parents in terms of chalk percentage: KBNT *lpa* has translucent grains containing 3.15% chalk, and ZHE733 with opaque grains contains 25.37% chalk [33]. The K/Z RIL population has previously been used for mapping biotic resistance, including water weevil and straight-head resistance [48]. 

### 2.2. Measurement of Grain Characteristics

The rice grains were dried to approximately 12% moisture with an air drier (Thermo Scientific Cytomat Automated Incubators, Watham, MA, USA) at 28 °C and then dehulled using an automatic rice husker (Kett, Villa Park, CA, USA). The percent chalk (PC), grain length (GL), grain weight (GW), and grain length to width ratio (RGLW) were calculated using WinSEEDLE Pro V.2007E (Regent Instruments Inc., Sainte-Foy, QC, Canada) with an Epson Perfection V800 photo scanner (Epson America Inc., Long Beach, CA, USA), using 100 g per line. Hundred grain weight (HGW) was calculated from the weight of 100 g (Mettler-Toledo, Greifensee, Switzerland). All measurements for each line were made with five replications per line and used for QTL analyses. 

### 2.3. Scanning Electron Microscopy

Rice grains of the two parental genotypes, KBNT *lpa* and ZHE733, were dehulled using an automatic rice husker (Kett, Villa Park, CA, USA) and fixed in 10% formalin for 24 hours. Fixed grains were rinsed with distilled water three times for 30 min. Rinsed grains were dehydrated using a standard series of ethanol: 25%, 50%, 75%, and 95% each for 20 min; then, they were washed with 100% ethanol three times for 30 min each. Grain samples were dried until the critical point. Dried grains were cut in half transversely using a scalpel, and halved grains were mounted on aluminum stubs with the fractures facing up. The specimens were gold coated at 35 mA for 120 s. The morphology of the endosperms was examined with a scanning electron microscope (Leo440 Stereoscan SEM, LEO/Leica, Cambridge, UK) at an accelerating voltage of 30.00 kV, a working distance of 10 mm, a probe current of 20 pA, and spot sizes of 500, 20, and 10 µm. SEM analysis was based on three biological replications per treatment. All of the SEM procedures were carried out according to the manufacturer’s protocol. 

### 2.4. Correlation of Grain Characteristics in the RIL Population

The Pearson correlation coefficients were calculated pairwise between the traits PC, GL, GW, RGLW, and HGW using JMP version 12.0.

### 2.5. Genotyping

Leaf samples of the two-week-old seedlings of the 198 selected lines and parents were used for genomic DNA extraction using the cetyl trimethyl ammonium bromide (CTAB) method, as described [49]. A single-end library with *Pstl* and *Mspl* enzymes was created for GBS by the University of Minnesota Genomics Center. Two million base pair (bp) reads were generated by NextSeq with a 1 × 150 bp read length. Reads with mean quality scores above Q30 were chosen for analysis. A FASTQ file was generated from the sequence reads for SNP identification. Illumina BCL2FASTQ software version 3.9. was used to de-multiplex the FASTQ files. The adapter sequences, comprising the first 12 bases of the individual reads, were removed using Trimmomatic version 0.32. Burrows–Wheeler Alignment (BWA) software version 0.7.17.tar.bz2 was used to align the FASTQ files to the reference genome of Nipponbare, *Oryza sativa* spp. japonica version MSU7. SNP calling was performed from the sequences that perfectly matched and aligned to the reference Nipponbare genome. The call variants across all samples were joined using Freebayes version 0.9.18, and a raw Variant Call Format (VCF) file was generated. The VCF file was filtered by using VCF tools to remove variants with a minor allele frequency of less than 1%, variants with genotype rates of less than 95%, samples with genotype rates of less than 50%, variants with 100% missing data, variants with monomorphic markers between parents, and variants with more than 50% heterozygosity. The file format of the identified SNPs was converted to an ABH-based format, where “A” is a donor allele, “B” is a recipient allele, and “H” is a heterozygous allele. 

### 2.6. Linkage Map Construction and QTL Analysis

A total of 4133 filtered SNP markers which covered 12 chromosomes were used to construct a linkage map using QTL IciMapping software version 4.2.53 [50] with the Kosambi mapping function [51], and with the recombination frequency (r) set at 0.45. The grain characteristics used to conduct QTL analysis were PC, GL, GW, RGLW, and HGW. A total of 198 lines were used for the QTL analysis. QTL analysis was performed using QTL IciMapping software version 4.2.53 with the inclusive composite interval mapping (ICIM) function. ICIM was used for additive, dominant, and epistatic QTL mapping in biparental populations, using background markers as cofactors for analysis. Significant QTLs were determined based on a phenotypic variance (PVE) ≥ 2% and a logarithm of odd (LOD) ≥ 2.5. QTL nomenclature followed that of Solis et al. (2018) [52] and McCouch (2008) [53] based on the trait name, chromosome number, and physical map position on the genome. 

### 2.7. Identification of Candidate Genes within the QTL Regions

The positions of the SNP markers flanking the QTL regions were used to identify the candidate genes present within the QTL regions. The MSU *japonica* rice reference genome annotation release 7.0 was used as the reference for physical mapping. 

## 3. Results

### 3.1. Distribution of Grain Characteristics in the RIL Population

The results from the measurements of five traits for the parents and the RIL population are summarized in Table 1. Significant differences for PC, GL, and RGLW were found between the two parents. ZHE733 had higher phenotypic values than KBNT *lpa* for PC and RGLW, but not for GL. Values for all the phenotypic traits in the RIL population showed continuous phenotypic variation with bell-shaped phenotypic distributions and a wide range of variation (Figure 1), suggesting that these traits were quantitatively inherited and suitable for genetic mapping QTL analysis. In the five measured traits, transgressive segregation in one or both directions was detected. The mean values of PC, GL, GW, RGLW, and HGW ranged from 3.05 to 73.22%, 5.56 to 8.08 mm, 1.86 to 3.01 mm, 0.25 to 0.48, and 1.28 to 3.19 g, respectively.

For PC, the values for the RIL population ranged from 3.05 to 73.22%, and KBNT *lpa* showed 3.15% while ZHE733 showed 35.15%. The frequency distribution of PC in the RIL population exhibited transgressive segregation for higher PC, with 36% of lines exceeding the PC of ZHE733 as the parent with higher chalkiness, whereas only 1.01% of lines showed lower PC than KBNT *lpa* as the parent with lower chalkiness (Figure 1A). Furthermore, based on the frequency distribution, Figure 1B–D demonstrate that most of the lines showed similarity with ZHE733 in terms of GL, GW, and RGLW. In addition, for HGW, most of the lines have a lower value than that of both parents (Figure 1E).

### 3.2. Morphology of Starch Granules

Based on the scanning electron microscopy (SEM) analysis, translucent grains (KBNT *lpa*) and chalky grains (ZHE733) showed significant differences in their starch granule structures (Figure 2). Translucent grains displayed a thick aleurone layer, compact with a high density of starch granules located in the central area (Figure 2A), whereas chalky grains exhibited a thin aleurone layer, loose with a low density of starch granules that were mostly located in the peripheral area (Figure 2B). Furthermore, the shape of the starch granules in the translucent grains showed a polygonal structure (Figure 2C), while the chalky grains had rounded starch granules with a larger space between them (Figure 2D). Starch granules of the chalky grains showed cavities (Figure 2F), while there were no cavities in the translucent grains (Figure 2E).

### 3.3. Correlation of Grain Characteristics in the RIL Population

The phenotypic correlation coefficients of grain characteristics are listed in Table 2. A strong positive correlation was observed between PC and GW (*r =* 0.56; *p* < 0.05), indicating that the broader the rice grain is, the chalkier the rice is. RGLW was strongly and positively correlated with PC (*r =* 0.33; *p* < 0.05), GL (*r =* 0.69; *p* < 0.05), and GW (*r =* 0.82; *p* < 0.05), and the grain size, GW, and RGLW had an important impact on PC. Furthermore, the correlation coefficient for HGW with each of the traits was small, indicating the complexity of the relationship among the traits. 

### 3.4. QTLs for Percent Chalk (PC)

A genetic map was constructed based on 198 lines of K/Z RIL populations and had a total length of 6063.12 cM with an average distance of 1.58 cM between adjacent markers, covering 373 Mb of the rice genome in total. This high-density genetic map was used to identify QTLs with higher resolution and to identify candidate genes more accurately [54]. Table 3 displays the significance of the logarithm of odds (LOD), percent of variation explained (PVE), peak markers, marker position, candidate genes, and gene annotation of all the significant QTLs for grain characteristics. 

Fifteen QTLs controlling percent chalk (PC) were identified and mapped to chromosomes 1, 2, 3, 4, 5, 6, 7, 9, 10, 11, and 12 (Table 3 and Figure 3) and tentatively named after the chromosome mapped for *qPC1.1*; *qPC2.1*; *qPC3.1*; *qPC4.1*, *qPC4.2*, *qPC4.3*, and *qPC4.4*; *qPC5.1* and *qPC5.2*; *qPC6.1*; *qPC7.1*; *qPC9.1*; *qPC10.1*; *qPC11.1*; and *qPC12.1*, respectively. All of the QTLs for PC showed a high value of LOD and PVE, ranging from 3.04 to 4.11 and 3.04 to 7.29%, respectively. Within the QTL regions, 49 candidate genes for PC were found, with gene annotation involved in many biological processes, molecular functions, and cell components (Table 3).

### 3.5. QTLs for Grain Length (GL)

Two QTLs (Table 3 and Figure 3) for GL were detected and mapped to chromosomes 11 and 12, named *qGL11.1* and *qGL12.1*, respectively. The QTL *qGL11.1* explained 8.44% of the variation with an LOD score of 4.96, located at 405 cM near the SNP marker KZ104292 on chromosome 11. Another QTL, *qGL12.1*, was identified at 263 cM near the SNP marker KZ105819 on chromosome 12, with 7.07% PVE and an LOD score of 2.81. Within the QTL regions, two candidate genes were found, with gene annotation predicted to be a b-ZIP transcription factor. 

### 3.6. QTLs for Grain Width (GW)

A total of four QTLs were found for GW (Table 3 and Figure 3), with one each on chromosomes 1, 6, 10, and 11, named *qGW1.1*, *qGW6.1*, *qGW10.1*, and *qGW11.1*, respectively. The range of the PVE for all these QTLs was 5.70–8.52%, and the LOD score range was 2.62–3.42. Within the QTL regions, 16 candidate genes were found, with gene annotations described as thiamin pyrophosphokinase 1, Hemoglobin Hb2, powdery mildew resistance protein, and WRKY gene 41.

### 3.7. QTLs for Hundred Grain Weight (HGW)

On chromosomes 3 and 6, QTLs for HGW were identified, named *qHGW3.1* and *qHGW6.1*, respectively. *qHGW3.1* is located on chromosome 3 at 507 cM near the SNP marker KZ35937, with an LOD score of 7.25, which explained 16.26% of the variation. The second QTL, *qHGW6.1*, is at 107 cM near the SNP marker KZ67895 on chromosome 6, which explained 6.79% of the variation, with an LOD score of 3.19 (Table 3 and Figure 3). Within the QTL regions, four candidate genes were found, with gene annotation associated with enoyl-CoA hydratase protein.

### 3.8. QTLs for Grain Length to Width Ratio (RGLW)

Five QTLs were detected for RGLW, with one on chromosome 3 (*qRGLW3.1*), two on chromosome 4 (*qRGLW4.1* and *qRGLW4.2*), and two on chromosome 11 (*qRGLW11.1* and *qRGLW11.2*), shown in Table 3 and Figure 3. Each of these QTLs explained 2.51–3.13% of the variation with LOD scores of 2.61–2.91. Within the QTL regions, 21 candidate genes were found, with gene annotation related to zinc finger protein.

## 4. Discussion

Recently, issues related to climate change are increasing the impact of abiotic and biotic stresses. An understanding of the genetic information associated with the phenotypes of rice grain chalk percent and structural characteristics is important to develop rice varieties with good quality and high yield. Molecular markers linked to chalkiness and rice grain characteristics can be used to eliminate poor or undesired alleles to improve quality and yield in rice breeding programs through marker-assisted selection (MAS).

In this study, a recombinant inbred line (RIL) population derived from the cross of the upland japonica variety Kaybonnet (KBNT *lpa*) and the indica variety ZHE733 (named K/Z RIL population) was used. The novelty of this study is related to the genetic composition of the RIL population, being a cross of different subspecies that contains much genetic recombination, and the QTLs were analyzed by using high-throughput SNP markers. Previously, many studies conducted QTL analyses associated with rice grain appearance by using many different kinds of mapping populations, including F2 populations [40]; a doubled-haploid (DH) population derived from a cross between the same subspecies or different subspecies [55]; a near-isogenic line (NIL) population [32,45]; recombinant inbred line (RIL) populations from crosses within the same subspecies, or between different subspecies or from elite hybrids [11,28,33,34,39,56,57,58]; a backcross population [59,60]; and a chromosomal segment substitution line (CSSL) population [29,35,61], and only used standard markers for QTL analysis. 

According to Edwards et al. (2017) [33], genetic and environmental factors influence the rice grain characteristics of segregating populations inheriting the complex rice grain quality traits. The parents of the K/Z RIL population, KBNT *lpa* and ZHE733, exhibited a significant difference in percent chalk (PC), grain length (GL), and grain length to width ratio (RGLW) (Table 1). The K/Z RIL population showed a continuous distribution for grain characteristics, suggesting that these phenotypes are controlled by quantitative trait loci (Figure 1). Transgressive segregation for single or both directions was detected in all the grain characteristics (PC, GL, GW, RGLW, and HGW). The direction of the transgressive segregation in PC towards increased PC (greater than the higher PC parent, ZHE733) could be influenced by the *lpa1-1* mutant allele [33]. The transgressive segregation for GL was more towards a reduction in GL (less than the shorter GL parent, ZHE733), while GW showed transgressive segregation towards an increase in GW (greater than the wider GW parent, ZHE733), and the direction for RGLW leaned more towards an increase in RGLW (more than the higher RGLW parent, ZHE733), whereas the transgressive segregation for HGW was more towards a reduction in HGW (less than both parents, KBNT *lpa* and ZHE733). This transgressive segregation may be due to epistasis or additive effects at multiple recombined loci, which may be because of the inter-subspecies cross between KBNT *lpa* as a tropical japonica and ZHE733 as an indica subspecies, which could influence the characteristics of PC, GL, GW, RGLW, and HGW in the K/Z RIL population [33]. Based on the results from a previous analysis, PC correlated with grain shape traits [62]. The results of this study are also consistent with previous results, with the grain size (GL, GW, and RGLW) loci showing strong positive correlations with PC (Table 2). 

The scanning electron microscopy (SEM) observation revealed that in KBNT, the *lpa* mutant displays a translucent grain that shows polygonal and compact starch granules, while ZHE733 has a chalky grain that displays loose and round starch granules. According to Lisle et al. (2000) [23], chalky grains have a lower starch concentration compared to translucent grains. Chalky grains contain less amylose and more amylopectin with long branched chains compared to translucent grains [63]. Chalky grains display an opaque appearance due to the spaces between the loosely packed starch granules that block light transmission. The spaces also generate mechanical breakage during the milling process [28], causing a reduction in grain quality. 

QTL mapping has frequently been used to detect QTLs and candidate genes within the QTL regions, and the most reliable molecular markers used have been single-nucleotide polymorphisms (SNPs) due to their broad distribution throughout the rice genome compared to conventional markers such as simple sequence repeat (SSR), restriction fragment length polymorphism (RFLP), and amplified fragment length polymorphism (AFLP) [64]. A high-density genetic linkage map can be generated by using higher number of SNPs, which thus allows higher efficiency of QTL mapping that can identify more precise QTLs and candidate genes [65]. In this study, high-throughput SNP markers were used. 

In this study, a total of twenty-eight QTLs and ninety-two candidate genes related to rice grain characteristics were identified (Table 3 and Figure 3). Some QTLs showed a pleiotropic or location-linked association that was influenced by the phenotypic correlations between rice grain characteristics. Two QTLs, for PC (*qPC6.1*) and GW (*qGW6.1*), located on chromosome 6 showed a pleiotropic or location-linked association since PC and GW exhibited a strong positive correlation (*r* = 0.56; *p* < 0.05) (Table 2). Fitzgerald et al. (2009) [4] also reported that PC was associated with GW. Chromosome 6 has been reported to be a grain quality QTLs hotspot associated with percent chalk, grain characteristics, amylose, head rice yield, gelatinization temperature, and gel consistency [66]. Furthermore, most of the QTLs were independent of one another as they were at different genome locations. 

According to a review paper by Sreenivasulu et al. (2015) [66], 140 QTLs associated with chalkiness were found on all 12 chromosomes of rice, and several genes were identified within those QTL regions, such as *WAXY* [67,68], chalk5 [36], starch synthase III A [26], pyruvate orthophosphate dikinase [24], cell wall invertase [69], UGPase1 (UDP-glucose pyrophosphorylase 1) [70], GIF1 (Grain Incomplete Filling 1) [69], and H^+-^translocating pyrophosphatases [36]. Overlapped QTLs from previous studies include *qGW1.1*, which overlapped with a QTL for GW on chromosome 1 that was identified by Tan et al. (2001) [56]. The QTL *qPC2.1* on chromosome 2 is located near the *lpa1-1* gene, consistent with the QTL for chalkiness reported by Edwards et al. (2017) [33]. Landoni et al. (2013) [71] suggested that the *lpa* gene influenced starch granule characteristics in maize. 

Three QTLs identified on chromosome 3, namely *qPC3.1*, *qRGLW3.1,* and *qHGW3.1*, have overlapping regions for chalkiness and grain characteristics, as reported by Tan et al. (2001) [56]. On chromosome 4, four QTLs associated with chalkiness (*qPC4.1*, *qPC4.2*, *qPC4.3*, and *qPC4.4*) were detected, and two QTLs related to ratio of grain length to grain width (*qRGLW4.1* and *qRGLW4.2*) were identified. These QTLs overlap with the *flo-2* locus (locus of rice floury endosperm) that regulates the expression of genes related to starch synthesis, including rice starch-branching enzyme (RBE) and granule-bound starch synthase (GBSS) [72]. Two QTLs associated with chalkiness (*qPC5.1* and *qPC5.2*) were detected on chromosome 5 that overlapped with the *flo-1* locus [73], Chalk5 [36], and a QTL related to percentage of grain with chalkiness (PGWC) [73]. The *flo-1* locus determines a floury white endosperm that shows round and loosely stacked starch granules [10]. Chalk5 is a major QTL for chalkiness, and overexpression of Chalk5 leads to the development of loosely deposited starch granules with spaces around the starch granules and protein bodies. Li et al. (2014) [36] suggested that overexpression of Chalk5 disturbs the pH homeostasis of the endomembrane trafficking system and changes the biogenesis of protein bodies in the endosperm development period. 

The QTL on chromosome 6 associated with chalkiness, *qPC6.1*, is close to the *WAXY* gene. Zhao et al. (2009) [42] also identified a QTL related to chalkiness on chromosome 6 close to the *WAXY* gene. According to Peng et al. (2014) [74], eight chalkiness QTLs have been detected near the *WAXY* gene. One chalkiness QTL was identified on chromosome 7, *qPC7.1*, which overlapped with the chalkiness QTLs detected by Zhao et al. (2009) [42]. On chromosome 9, *qPC9.1* was detected close to the UDP-glucose pyrophosphorylase1 (*UGPase1*) gene, which plays an important role in the development of chalky endosperm [70]. Stable chalkiness QTLs across many environments were fine mapped onto chromosome 9, also close to *UGPase1* [58]. One chalkiness QTL identified on chromosome 10, *qPC10.1,* co-localized with a chalkiness QTL detected by Liu et al. (2012) [73]. On chromosome 11, five QTLs were identified, namely *qPC11.1, qGL11.1, qGW11.1, qRGLW11.1*, and *qRGLW11.2*; *qGW11.1* was co-located with the grain width QTL detected by Tan et al. (2001) [56]. Two QTLs were detected on chromosome 12: *qPC12.1* and *qGL12.1*. Grain chalkiness is a highly undesirable trait in the market. Based on this QTL information, marker-assisted selection (MAS) can be used to eliminate alleles for high chalkiness in rice breeding programs [4,66,75]. 

## 5. Conclusions

Considering the increasing concern in the rice industry regarding the eating and milling quality of rice, the identification of genetic information related to grain characteristics is important to provide a strategy for improving rice quality. Continuous distributions and transgressive segregations of all the traits were observed, suggesting that the traits were quantitatively inherited. A total of twenty-eight QTLs and ninety-two candidate genes related to rice grain characteristics were identified. The QTL for chalkiness on chromosome 2, *qPC2.1*, located near the *lpa1-*1 gene, might be the cause of chalkiness in this K/Z RIL population. Two QTLs for PC (*qPC6.1*) and GW (*qGW6.1*), located on chromosome 6, showed a pleiotropic or location-linked association, as PC and GW exhibited a strong positive correlation (*r =* 0.56; *p* < 0.05). The scanning electron microscopy (SEM) observation revealed that there was a significant difference between KBNT *lpa* and ZHE733 in terms of the starch granules’ morphology. KBNT *lpa* as a translucent grain showed polygonal and compact starch granules, while ZHE733 as a chalky grain displayed loose and round starch granules. Hence, this study provides valuable information to develop rice varieties of high quality. In future studies, it will be important to clone the genes within the QTLs *qPC2.1*, *qPC6.1*, and *qGW6.1* to validate the function of the genes in chalkiness and grain characteristics. 

## Figures and Tables

**Figure 1 genes-12-01690-f001:**
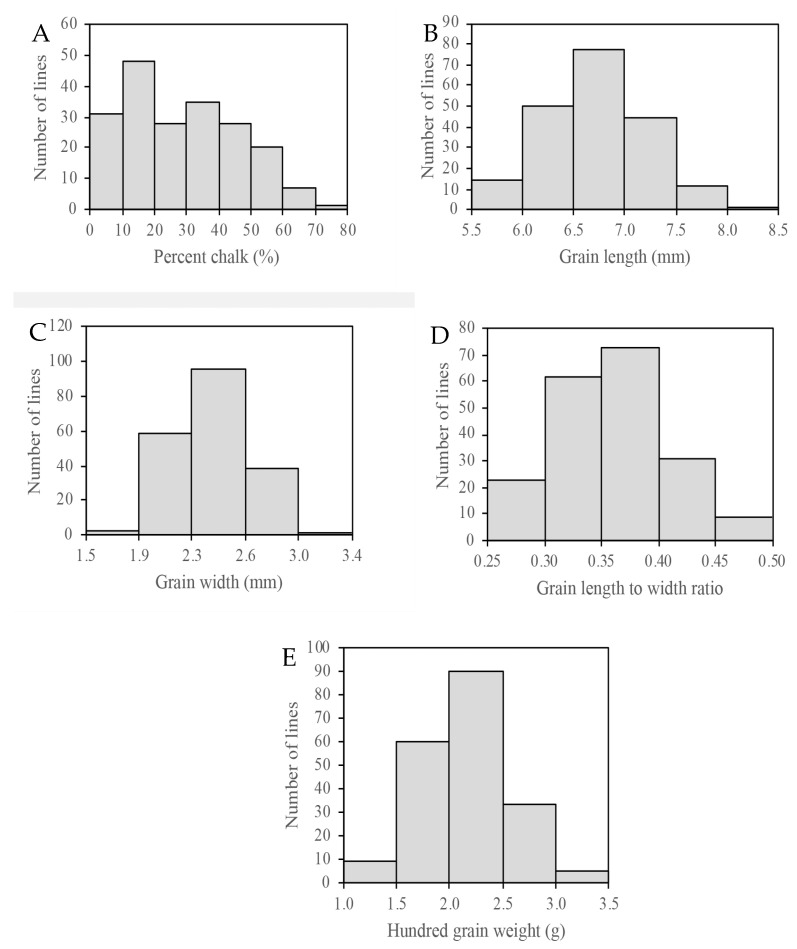
Frequency distribution of number of lines with percent chalk (**A**), grain length (**B**), grain width (**C**), grain length to width ratio (**D**), and hundred grain weight (**E**) of the K/Z (KB x ZHE733) RIL rice population. Abbreviations: Kaybonnet KB/KBNT, *lpa*; Zhe, ZHE733.

**Figure 2 genes-12-01690-f002:**
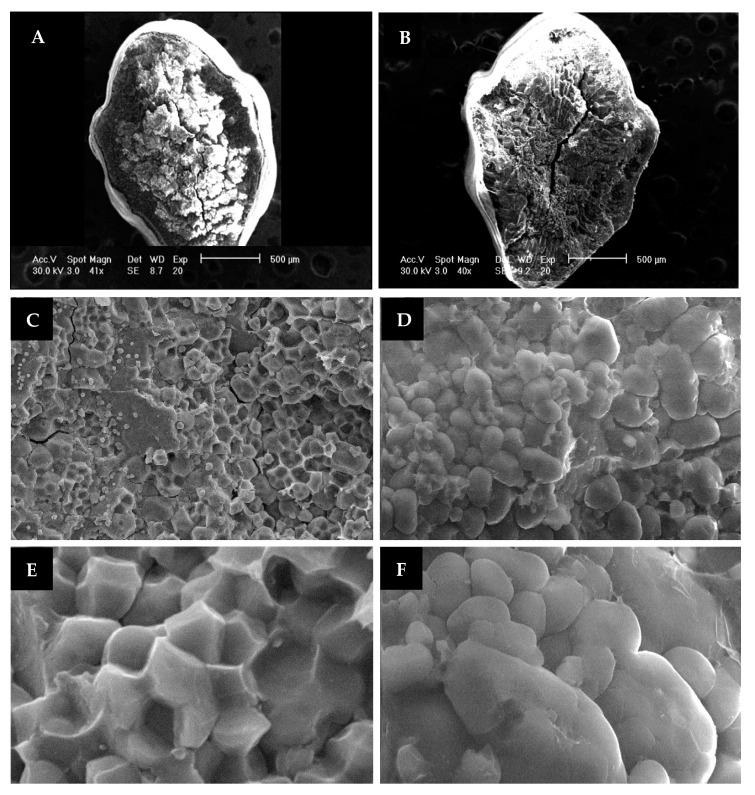
Transverse cross-section images of mature rice grain with Scanning Electron Microscope (SEM). (**A**,**B**) at 40× magnification; (**C**,**D**) at 1000× magnification; (**E**,**F**) at 5000× magnification. (**A**) *O. sativa* cv. KBNT *lpa* as parent of K/Z RIL population showing translucent grain; (**B**) *O. sativa* cv. ZHE733 as parent 2 of K/Z RIL population showing chalky grain; (**C**) *O. sativa* cv KBNT *lpa* at 1000× mag-nification, showing KBNT *lpa* parent showing translucent grain; (**D**) parent ZHE733 showing chalky grain; (**E**) KBNT *lpa* parent showing translucent grain, and (**F**) *O.sativa* cv ZHE733 parent showing chalky grain.

**Figure 3 genes-12-01690-f003:**
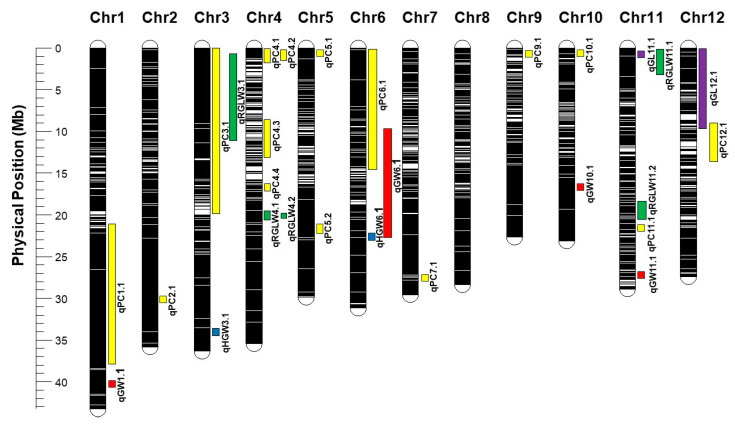
Locations of the QTLs for percent chalk (PC), grain length (GL), grain weight (GW), grain length to width ratio (RGLW), and hundred grain weight (HGW) of K/Z RIL population.

**Table 1 genes-12-01690-t001:** Phenotypic values of percent chalk, grain length, grain width, grain length to width ratio, and hundred grain weight of the K/Z RIL population.

Trait ^1^	KBNT *lpa* (Mean ± SD)	ZHE733 (Mean ± SD)	K/Z RIL Population	
			(Mean ± SD)	Range
PC (%)	3.15 ± 0.14a ^2^	35.15 ± 0.57b	28.54 ± 2.47	3.05–73.22
GL (mm)	7.28 ± 0.05a	6.52 ± 0.03b	6.73 ± 0.07	5.56–8.08
GW (mm)	2.12 ± 0.01a	2.54 ± 0.04a	2.39 ± 0.03	1.86–3.01
RGLW	0.29 ± 0.00a	0.38 ± 0.00b	0.36 ± 0.01	0.25–0.48
HGW (g)	2.55 ± 0.01a	2.63 ± 0.01a	2.14 ± 0.07	1.28–3.19

^1^ PC, percent chalk; GL, grain length; GW, grain width; RGLW, grain length to width ratio; HGW, hundred grain weight. ^2^ Different letters (a and b) indicate a significant difference between the two parental lines (*p <* 0.05); traits with the same letter are not significantly different.

**Table 2 genes-12-01690-t002:** Phenotypic correlations between the traits of percent chalk, grain length, grain width, grain length to width ratio, and hundred grain weight of K/Z RIL population.

	PC	GL	GW	RGLW
GL	0.11			
GW	0.56 *	−0.18 *		
RGLW	0.33 *	0.69 *	0.82 *	
HGW	0.06	0.06	0.05	0.08

* Significant *p* < 0.05. PC, percent chalk; GL, grain length; GW, grain width; RGLW, grain length to width ratio; HGW, hundred grain weight.

**Table 3 genes-12-01690-t003:** QTLs identified for chalkiness and grain characteristics of K/Z RIL population with QTL IciMapping software version 4.2.53.

Trait Name	QTLs	Chro-Mo-Some	LOD	PVE (%)	Peak Marker	Positi-on (cM)	Candidate Gene Position (bp)	Number of Genes within a 25 kb Interval of the Marker Closest to the QTL Peak	Gene Annotation
PC	qPC1.1	1	4.11	7.29	KZ12316	594	20880897	4	Transposon protein
	qPC2.1	2	3.06	5.45	KZ24375	565	30984762	5	Matrix metalloproteinase
	qPC3.1	3	3.68	6.88	KZ34590	269	20457764	4	Transposon protein
	qPC4.1	4	3.67	6.13	KZ45081	132	3411195	4	Gene MEG family precursor
	qPC4.2	4	3.61	6.09	KZ45056	146	2031785	2	Protein kinase
	qPC4.3	4	3.6	6.11	KZ44765	166	14885436	2	ulp1 protease family
	qPC4.4	4	3.04	6.36	KZ44529	442	1743647	2	Similar to Subtilase
	qPC5.1	5	3.25	3.87	KZ55580	139	356032	4	Lysine ketoglutarate reductase
	qPC5.2	5	3.08	3.47	KZ56699	237	22985131	2	Thiamine pyrophosphate enzyme
	qPC6.1	6	3.59	6.07	KZ67354	39	17834132	4	ABC-2 type protein
	qPC7.1	7	3.32	3.04	KZ79405	300	28879166	2	Calmodulin protein kinases
	qPC9.1	9	3.96	3.27	KZ100068	106	1182883	3	Nodulation receptor kinase
	qPC10.1	10	3.33	3.05	KZ101986	274	1449944	4	CAF1 family ribonuclease
	qPC11.1	11	3.26	3.89	KZ104730	198	24102991	3	Retrotransposon protein
	qPC12.1	12	3.53	3.23	KZ105777	394	10678981	4	Retrotransposon protein
GL	qGL11.1	11	4.96	8.44	KZ104292	405	2464998	1	b-ZIP transcription factor 79
	qGL12.1	12	2.81	7.07	KZ105819	263	12920333	1	Hypothetical protein
GW	qGW1.1	1	2.62	5.7	KZ14572	20	40865364	5	Thiamin pyrophosphokinase 1
	qGW6.1	6	3.42	8.52	KZ67890	32	23243264	5	Similar to Hemoglobin Hb2
	qGW10.1	10	2.81	7.06	KZ102022	272	1917072	3	Powdery mildew resistance protein
	qGW11.1	11	3.07	7.21	KZ105089	275	27785210	3	WRKY gene 41
RGLW	qRGLW3.1	3	2.67	2.51	KZ33510	169	1393131	4	Homeobox domain protein
	qRGLW4.1	4	2.61	2.81	KZ45357	76	21827001	4	Zinc finger protein 28
	qRGLW4.2	4	2.91	2.77	KZ45359	390	21827105	4	Zinc finger protein 28
	qRGLW11.1	11	2.62	3.13	KZ104416	120	3740185	5	Zinc finger protein
	qRGLW11.2	11	2.79	2.65	KZ104272	407	2255547	4	PHD finger protein 42
HGW	qHGW3.1	3	7.25	16.26	KZ35937	507	35611464	2	Expressed protein
	qHGW6.1	6	3.19	6.79	KZ67895	107	23359587	2	Enoyl-CoA hydratase protein

## Data Availability

The data presented in this study are available on request from the corresponding author.

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
