# Peer review of "Identification of Genomic Regions Controlling Chalkiness and Grain Characteristics in a Recombinant Inbred Line Rice Population Based on High-Throughput SNP Markers"

_genes, 2021, doi:10.3390/genes12111690_

Round 1

Reviewer 1 Report

I have only a few suggestions here:

  1. A few grammar mistakes:    For example:  line 39, Slender and long grains that having a RGLW of three or above are generally preferred by the majority of rice consumers in most Asian countries. Please change it to either "grains that have a RGLW of three or above" or "grains having a RGLW of three or above".
  2. In Materials and Methods section, the authors did mention the parents and there are 198 lines in the RIL population. However, there are no exact number of samples used of each line for the experiments.  Additionally, as to the two parents, how many samples were used for each experiments  and data analysis? Please clarify this information. 
  3. For QTL analysis, please clarify the sample number.
  4. In figure 1, it is very confusing to read the figure because of the labeling of KB and Zhe on the top of the chart. I suggest the authors also put the K/Z RIL information on the graph so that it could be clear that the data points are collected from all of these three types of samples.
  5. In the Conclusion section, I suggest the authors write more about the future application of their discovery and also write more about the future experimental plans in order to further clarify the mechanisms behind these QTL discovery and phenotypes.

Reviewer 2 Report

The authors identified many QTLs and candidate genes controlling rice chalkiness and grain characteristics using a recombinant population with high density genetic map. In general, the manuscript is in good shape, following the general procedures of identifying genomic regions for traits. however, I think the manuscript can be improved if following points are taken into consideration, besides the grammar correction through the whole manuscripts.

1) In introduction, in P2-P3, line 74-108, the authors are trying to introduce the QTL/candidate genes identified for rice chalkiness and grain characteristics in literature. These two paragraphs for me are just piling things up, there is no organization somehow. The authors jumped from talking one trait then to another, then back to the trait again. they can try to either talk about traits one by one, or talk about the candidate gene by chromosome.  

In addition, following the objectives for this study, the authors should have a few sentences talking about what you have done to achieve the goals.

some other minor edits:

line 62-63: 'long grain rice varieties are dominant that susceptible to breakage due to chalkiness', this is not a correct sentence to me. I believe the authors are trying to say 'long grain rice varieties are dominant but they are susceptible to breakage due to chalkiness'?

line 70-73, line 97-100, grammar correction needed.

2) In M&M, I think section 2.7 should be moved up after 2.2 because they are phenotypes described, although no QTL analysis was conducted for this trait. accordingly, the results (in Results section 3.8) for this morphology of starch granule should match up with M&M. 

More importantly, the authors did the SEM only on the two parents as they presented in the results section, but they never mentioned this in M&M to my knowledge. 

in addition, in section 2.5, the authors mentioned the QTL analyses which were done using inclusive composite interval mapping (ICIM). What is ICIM? what is the model used in this method? the authors should describe these briefly.

3) In Results, P6, line 220-222, the authors state that 'grain size (GL, GW, and RGLW) had an important impact on PC, and also it is reasonable because RGLW usually increases with GL and GW'. firstly, from the table 2, GL is not correlated with PC. second, what does this sentence mean, 'it is reasonable because RGLW usually increases with GL and GW'? there is no reference for this statement, and also, RGLW is the ration between GL and GW, How could it increase with GL and GW if RGLW = GL/GW?

in Table 3, the authors consider the candidate genes within 25kb of the peak QTL. why choose 25kb of the peak QTL? usually, you can get the confidence interval around a peak QTL by dropping certain number of LOD value and getting the flanking marker. I am not sure why the authors chose 25kb.

In addition, grammar corrections are needed for the Results section. For instance, line 234, change 'led' to 'lead'; line 241, change 'number of ' to 'value of' or just remove 'number of'. line 243, change 'we' to 'were' ;line 277, 'explaining ' to 'explained'. there are some other ones, like line 288-289. also, the authors should keep using on tense across the whole section like they used past tense for most of the reports. But there are some places they mixed used different tenses. line 286-287, line 263 etc. 

4) In Discussion, the authors pile up a number of references to state that ' many studies conducted QTL analysis associated with the rice grain appearance by using many different kind of mapping population'. I am pretty sure there are a lot of studies about this using different rice mapping populations, but what are the novelty about this study? what is the difference between your study and the others? should not just pile the references up. 

line 320-321, the authors stated that 'Genetic and environmental factors influence the percent chalk and rice grain characteristics as the complex traits.' however, through this whole paragraph, there is nothing about the environmental factors affecting chalkiness and grain characteristics reported.

line 334, add 'may be' in front of 'due to'. line 339, add 'are' in front of 'also'.

line 348-353, this is just a repeat of what you already reported previously. the same for line 355-362, it is repeated.

line 402-405: 'On chromosome 9, qPC9.1 was detected close to UDP-glucose pyrophosphorylase1 (UGPase1) that play an important role in the development of a chalky endosperm [69]. The fine-mapped of stable chalkiness QTLs across many environments were mapped on chromosome 9 [58]. ' first, change 'play' to 'plays'. second, what is the message here? are the fine-mapped chalkiness QTLs across many environments on chromosome 9 the same as UGPase 1? or are they even related or how?

Generally speaking, the discussion section repeated a lot of the statements from the Results section, and lack of comparisons or statements of uniqueness of the study the authors did.
